# Effect of Carrier Lipophilicity and Preparation Method on the Properties of Andrographolide–Solid Dispersion

**DOI:** 10.3390/pharmaceutics11020074

**Published:** 2019-02-10

**Authors:** Guowei Zhao, Qingyun Zeng, Shoude Zhang, Youquan Zhong, Changhao Wang, Yashao Chen, Liquan Ou, Zhenggen Liao

**Affiliations:** 1Key Laboratory of Applied Surface and Colloid Chemistry, Ministry of Education, School of Chemistry and Chemical Engineering, Shaanxi Normal University, Xi’an 710119, China; weiweihaoyunqi@163.com (G.Z.); changhaowang@snnu.edu.cn (C.W.); 2Key Laboratory of Modern Preparation of Traditional Chinese Medicine, Ministry of Education, Jiangxi University of Traditional Chinese Medicine, Nanchang 330004, China; qy9410zeng@163.com (Q.Z.); zg6748426@163.com (S.Z.); ouliquan918@163.com (L.O.); lyzlyg@163.com (Z.L.); 3School of Pharmacy, Jiangxi University of Traditional Chinese Medicine, Nanchang 330004, China; zhongyouquan1994@163.com

**Keywords:** andrographolide, solid dispersion, carrier lipophilicity, preparation method

## Abstract

Solid dispersion (SD) is a useful approach to improve the dissolution rate and bioavailability of poorly water-soluble drugs. This work investigated the effects of carrier material lipophilicity and preparation method on the properties of andrographolide (AG)–SD. The SDs of AG and the carrier materials, polyethylene glycol (PEG) and PEG grafted with carbon chains of different length (grafted PEG), have been prepared by spray-drying and vacuum-drying methods. In AG–SDs prepared by the different preparation methods with the same polymer as carrier material, the intermolecular interaction, 5% weight-loss temperature, the melting temperature (*T*_m_), surface morphology, crystallinity, and dissolution behavior have significant differences. In the AG–SDs prepared by the same spray-drying method with different grafted PEG as carrier material, *T*_m_, surface morphology, crystallinity, and dissolution behavior had little difference. In the AG–SDs prepared by the same vacuum-drying method with different grafted PEG as carrier material, the crystallinity and *T*_m_ decreased, and the dissolution rate of AG increased with the increase of grafted PEG lipophilicity. The preparation method has an important effect on the properties of SD. The increase of carrier material lipophilicity is beneficial to the thermal stability of SD, the decrease of crystallinity and the increase of dissolution rate of a poorly water-soluble drug in the SD.

## 1. Introduction

In the biopharmaceutics classification system, poorly water-soluble and highly permeable compounds mostly belong to Class II. These drugs are poorly soluble with either slow or limited release resulting in solubility or dissolution rate-limited absorption. Therefore, these drugs have high permeability, but their oral bioavailability is generally low [1,2,3]. Solid dispersion (SD) is one of the most effective approaches to improve the solubility, dissolution rate, and the bioavailability of poorly water-soluble drugs [4,5,6,7]. In SD, the drug can be dispersed as separate molecules, amorphous particles, or crystalline particles, which can reduce the energy barrier associated with dissolution. Moreover, the interaction between drug and carrier can act to inhibit agglomeration or crystallization. In addition, the water-soluble carrier can improve the wettability of the drug, facilitate release of the drug in a supersaturated state, and so on. All of these properties of SD can promote the absorption of poorly water-soluble drugs. However, the SD carrier material is usually a hydrophilic polymer, such as polyethylene glycol (PEG), polyvinylpyrrolidone (PVP), hydroxypropylmethylcellulose (HPMC), hydroxypropyl methylcellulose acetate succinate (HPMCAS), and poly(vinylpyrrolidone-co-vinyl acetate) (PVPVA) [8,9,10,11,12,13,14], while the Class II drug is usually a hydrophobic compound. Therefore, the hydrophobic drug has very limited solubility in the carrier material, is often in a supersaturated state, and has a tendency to recrystallize in the preparation process (cooling or solvent removal) and during storage of SD [15,16,17,18]. 

PEG is widely used in SD owing to its low melting point and excellent solubility in water or organic solvents. In addition, PEG is suitable for the preparation of SD by the melting method and the solvent method [19,20]. In this work the carbon chains were grafted and fixed on PEG without changing the main structure of the carrier material, thereby increasing the lipophilicity of the hydrophilic carrier material. Increased carrier material lipophilicity can not only increase the interaction between the hydrophobic drug molecule and the carrier, but can also increase the thermal stability of the SD. At the same time, increasing the lipophilicity of the carrier material can improve the compatibility between carrier material and hydrophobic drug, facilitating both the solubility and dissolution rate of the poorly water-soluble drug.

Andrographolide (AG), a diterpenoid lactone, is isolated from the Chinese herb *Andrographis paniculata*. It is known as the ‘king of bitters’ and has been proved to have many pharmacological actions, such as analgesic, antipyretic, anti-inflammatory, anti-infection, antiviral, anticancer, anti-hyperglycemia, anti-angiogenesis, immunostimulation, hepato-protection, antifertility, and anti-HIV effects. The potential use of AG has attracted wide attention in recent years. AG, a natural lipophilic molecule, has high lipophilicity (log P = 2.63) and low aqueous solubility (74 μg/mL). The therapeutic use of AG is restricted by its poor solubility in water, which results in low bioavailability after oral administration [21,22,23,24,25,26,27,28,29]. Therefore, AG was chosen as the model drug in this work. 

In this work, AG–SD was prepared by two industrially scalable technologies (spray-drying method and vacuum-drying method) [30,31,32,33], using PEG grafted with carbon chains of different length (grafted PEG) and PEG as carriers, and further characterized for the intermolecular interactions, thermal properties, surface morphology, specific surface area and pore volume of the SDs, and crystallinity and dissolution behavior of AG in the SDs. The effects of carrier lipophilicity and preparation method on the physicochemical properties and the dissolution behavior of AG–SD were investigated.

## 2. Materials and Methods

### 2.1. Materials

AG, PEG4000, and PEG8000 were purchased from Hao-Xuan Biotechnology Co., Ltd. (Xi’an, China), Guangfu Institute of Fine Chemical Industry (Tianjin, China), and MP Biomedical Co., Ltd. (Santa Ana, CA, USA), respectively. The grafted PEGs were synthesized in the laboratory, including PEG4000 laurate (SM1), PEG4000 palmitate (SM2), PEG4000 behenate (SM3), PEG8000 laurate (SM4), PEG8000 palmitate (SM5), and PEG8000 behenate (SM6). 

### 2.2. Preparation of Physical Mixture (PM)

AG and the carrier, with mass ratio of 1:3, were ground together using a pestle and mortar [34]. The mixture was sieved by a standard sieve (60 mesh) to yield a PM. The prepared PMs were denoted as AG–PEG4000–PM, AG–PEG8000–PM, AG–SM1–PM, AG–SM2–PM, AG–SM3–PM, AG–SM4–PM, AG–SM5–PM, and AG–SM6–PM, respectively. The samples were stored in a vacuum desiccator over silica gel at 25 °C (0% RH) for further examination.

### 2.3. Preparation of AG–SD by Spray-Drying Method (S)

The above PM was dissolved in 80% ethanol. The solution was spray-drying using a B-290 mini spray dryer (Büchi, Flawil, Switzerland). The spray-dryer inlet temperature was set at 50 °C, the pump rate was 2 mL/min, and the aspirator was set at 100% and the N_2_ flow at 40 m^3^/h. The powder was collected and stored in a vacuum desiccator over silica gel at 25 °C (0% Relative Humidity) for characterization. The prepared SDs were denoted as AG–PEG4000–S–SD, AG–PEG8000–S–SD, AG–SM1–S–SD, AG–SM2–S–SD, AG–SM3–S–SD, AG–SM4–S–SD, AG–SM5–S–SD, and AG–SM6–S–SD, respectively.

### 2.4. Preparation of AG–SD by Vacuum-Drying Method (V)

The above PM were dissolved in 80% ethanol and evaporated by an OSB-2000 rotary evaporator (Ailang Instrument Co. Ltd., Shanghai, China) over a water bath (80 °C) in a vacuum; the remainder was crushed into powder using a pestle and mortar and sieved by a standard sieve (60 mesh), and then stored in a vacuumed desiccator over silica gel at 25 °C (0% RH) for characterization. The prepared SDs were denoted as AG–PEG4000–V–SD, AG–PEG8000–V–SD, AG–SM1–V–SD, AG–SM2–V–SD, AG–SM3–V–SD, AG–SM4–V–SD, AG–SM5–V–SD, and AG–SM6–V–SD, respectively.

### 2.5. Fourier Transform Infrared Spectroscopy (FT-IR)

FT-IR spectra were obtained using a Spectrum Two FT-IR spectrometer (PerkinElmer Corp., Waltham, MA, USA). About 2–3 mg of the sample were mixed with dry KBr. The powder was compressed in a hydraulic press to form a disc by a powder compressing instrument (FW-4A, Uncommon, Tianjin, China) for FT-IR analysis. The spectra of the sample were scanned over a frequency range 4000–400 cm^−1^ with a resolution of 4 cm^−1^.

### 2.6. Thermogravimetric Analysis

Thermogravimetric (TG) analysis was carried out using a TG/DTA6300 thermal analysis instrument (SII Nano Technology Inc., Tokyo, Japan). Approximately 5 mg samples were placed in aluminum pans and heated from 30 to 500 °C with a heating rate of 10 °C/min. 

### 2.7. Differential Scanning Calorimetry

Differential scanning calorimetry (DSC) curves were obtained by a Diamond DSC instrument (PerkinElmer Corp., Waltham, MA, USA). Calibration of the DSC instrument was carried out using indium as a standard. Samples of 5 mg were loaded in aluminum pans and placed into DSC cells. Thermal analysis of samples was carried out at a scanning rate of 10 °C/min in the purge gas of nitrogen, over a temperature range of 20–270 °C.

### 2.8. Powder X-ray Diffraction

Powder X-ray diffraction (XRD) patterns were collected on a D8 ADVANCE-D8X X-ray diffraction (Bruker AXS GMBH, Karlsruhe, Germany) with a Cu Kα line as the source of radiation (*λ* = 1.541 Å). Standard runs were carried out using a voltage of 40 kV, a current of 40 mA, and a scanning rate of 8 °/min over a 2*θ* range of 5°–55° with a step size of 0.02°. The same sample of alpha-alumina as an external standard was also scanned in order to correct for the fluctuations in detector responses.

### 2.9. Scanning Electron Microscopy (SEM)

SEM images were recorded on a Quanta 250 scanning electron microscope (FEI Corp., Hillsboro, OR, USA). The samples were mounted on an aluminum stub with double-sided adhesive tape and coated under vacuum with gold in an argon atmosphere prior to the observation. 

### 2.10. Specific Surface Area and Pore Volume

The specific surface area and pore volume were determined by nitrogen gas absorption based on the Brunauer-Emmett-Teller method [35] using a TriStar3000 surface area and pore volume analyzer (Micromeritics Instrument Corp., Atlanta, GA, USA). The amount of nitrogen adsorbed was measured at partial nitrogen vapor pressure (*p*/*p*_0_) ranging between 0.05 and 0.35. Before measurement, the samples were performed with a continuous nitrogen flow at room temperature overnight to purge out the moisture.

### 2.11. Particle Size

Particle size was measured using a laser diffraction particle size analyzer (Mastersizer 2000, Malvern, UK). The intake air pressure and feed-rate of the operating parameters of the experiment were 2.5 bars and 55%, respectively. 

### 2.12. High Performance Liquid Chromatography (HPLC) Analysis

The content of AG was determined using an appropriate HPLC method. The analysis was performed using a 1260 HPLC system (Agilent Corp., Palo Arto, CA, USA). The column was Yilite C18 (150 mm × 4.6 mm, 5 μm). The mobile phases were methanol and water (60:40, *v*:*v*), the flow rate was 1 mL/min, the column temperature was 30 °C, and the wavelength of the UV detector was 225 nm. This method for determination of AG was validated by the methodological study in the preliminary experiment.

### 2.13. Dissolution Testing

The dissolution testing was tested using the Pharmacopoeia of China (Chinese Pharmacopoeia Commission, 2015) type 2 dissolution testing apparatus (paddle method). A ZRS-8G dissolution tester (Tianda-Tianfa Technology Co., Ltd., Tianjin, China) was used in this study. The samples were accurately weighed (0.10 g) and put into the vessels with 900 mL double distilled water (*n* = 6); paddle speed was 100 rpm and temperature was 37 ± 0.5 °C. The dissolution process was monitored for 2 h and the 1.5 mL samples were taken at 5, 10, 15, 30, 45, 60, 90, and 120 min and replaced with an equal volume of the same fresh medium. An aliquot of 1.5 mL was filtered through a 0.22 μm filter, and the concentration of AG was determined according to the above-mentioned HPLC condition.

## 3. Results

### 3.1. Investigation of Drug–Carrier Interactions by FT-IR

IR is a well-established method for characterizing intermolecular interactions, such as H-bonding, and has been extensively applied to probe the drug-carrier interactions in SD [6]. Figure 1, Figure 2, and Table 1 show the structures of AG and carriers, the FT-IR spectra of some samples, and the –OH peak position of AG, respectively. 

Table 1 showed the peak position at 3317 cm^−1^, corresponding to –OH vibrations of AG, where the blue shift in peak position were obviously observed upon mixing with carriers as SDs. This may be attributed to a change in the groups that form hydrogen bonds. The formation of intermolecular hydrogen bonds in the pure AG originates from the interaction between the –OH of the two AG molecules. While SDs were prepared, the interaction between the –OH of AG and the –C=O of grafted PEG or the –OH of PEG resulted in the formation of intermolecular hydrogen bonds. The degree of –OH blue shift of AG in AG-grafted PEG–S–SD was higher than that of AG in AG–PEG–S–SD, which was ascribed to the stronger interaction between –C=O of grafted PEG and –OH of AG than that between –OH of PEG and –OH of AG. The FT-IR spectra showed the original AG intermolecular hydrogen bond was destroyed in the as-prepared SDs, which was beneficial to the dissolution of AG. At the same time, the formation of an intermolecular hydrogen bond between AG molecule and carrier was beneficial to the SD stability [6,36]. 

### 3.2. Investigation of Thermal Stability by TG

TG analysis can be used to compare the thermal stability of SD and PM [37,38,39]. The TG curves of some samples and 5% weight-loss temperature (*T*_i_) of all the samples are shown in Figure 3 and Table 2, respectively. As shown in Table 2, compared with the SDs and the PMs prepared with grafted PEG as carrier, the *T*_i_ of SDs and PMs prepared with PEG as carrier was higher. The reason was that the grafted PEG contained ester bonds and some of the ester bonds were broken during the heating, resulting in the decrease of *T*_i_ of the SDs and PMs prepared with grafted PEG as carrier material.

From SM1 to SM5, the *T*_i_ of SDs prepared by spray-drying method and PMs increased gradually, suggesting that the intermolecular interactions between grafted PEG and AG enhanced with increasing grafted PEG lipophilicity. However, no obvious advantage of *T*_i_ for the SD was observed compared with that of the corresponding PM. This may be ascribed to the grafted PEG and AG in the PM melted and formed SD during the heating of measuring TG curve. 

### 3.3. Solid State Characterization by DSC

Figure 4 shows the DSC thermograms of some samples. As shown in Figure 4, the DSC thermogram of pure AG was typical of a crystalline substance with a sharp endothermic peak at 243 °C corresponding to its melting point. Table 2 shows the melting temperature (*T*_m_) of all the samples near 190–250 °C. Compared with the SDs prepared with grafted PEG as carrier, the *T*_m_ of the SDs prepared with PEG as carrier was lower. Since some of the ester bonds of the grafted PEG were broken during heating, both the compatibility of the grafted PEG and AG as well as the solubility of AG in the grafted PEG were reduced, leading to the increase of *T*_m_ of the SDs prepared with grafted PEG as carrier material. In all the SDs, a small endothermic peak was observed at about 220 °C (Figure 4 and Table 2). This result indicated that the completely amorphous AG–SD had not been obtained. AG was in a partial-amorphous and partial-crystal state in the as-prepared SDs. However, the glass transition was not observed on the DSC curve of the as-prepared SDs. This was due to the carrier material melting earlier, allowing the amorphous AG to dissolve in earlier melted carrier material during the heating of the DSC studies.

### 3.4. Solid State Characterization by XRD

XRD [40] is the most convenient method for evaluating the structure and crystallinity. XRD patterns of some samples are shown in Figure 5. The XRD pattern of pure AG had sharp peaks at 2*θ* 9.78°, 11.97°, 14.78°, 15.67°, 17.67°, 18.44°, and 22.62°, suggesting a typical crystalline structure. 

The characteristic diffraction peaks of crystal AG were present in all PMs (Figure 5). Compared with PMs, the intensity of AG diffraction peaks decreased significantly in the SDs. The AG diffraction peaks at 9.8°, 12.0°, and 15.7° were not disturbed by the carriers, so their peak areas were chosen to estimate the relative crystallinity of AG in the SD. The peak areas at 9.8°, 12.0°, and 15.7° in the SD prepared by spray-drying method and the SD prepared by the vacuum-drying method (except AG–SM6–V–SD) were about 30% and 60–70% of the corresponding PM, respectively. The crystallinity of AG in the SD prepared by spray-drying method was lower than that in the SD prepared by vacuum-drying method. The reason is that the ethanol solution of AG and carrier was rapidly evaporated, and the AG mostly solidified in an amorphous state during the process of SD prepared by spray-drying method. The solvent evaporation rate was slower, and the recrystallization of AG occurred during the process of SD prepared by vacuum-drying method, leading to the higher crystallinity of AG in the SD prepared by vacuum-drying method (except AG–SM1–V–SD and AG–SM6–V–SD). 

### 3.5. Morphological Evaluation

SEM was used in order to determine the surface morphology of the samples. As shown in Figure 6, the SEM images showed that the PMs (except AG–PEG8000–PM) and the SDs prepared by vacuum-drying method displayed similar irregular block-shaped particles with rough surfaces, because these powders were obtained using the same process of grinding and sieving. The original surface morphology of AG and PEG8000 could be observed in AG–PEG8000–PM. The SDs prepared by spray-drying method consisted of rod-shaped particles with irregular projections. The SEM images show that the SDs prepared by the same method had similar surface morphology.

### 3.6. Specific Surface Area, Pore Volume, and Particle Size

Table 3 listed the specific surface area, pore volume, particle size, and span of all the SDs. Compared with the SDs prepared by vacuum-drying method, the particle size of the SDs prepared by spray-drying method was smaller and the distribution was more uniform. However, the SDs prepared by spray-drying method had smooth surfaces, compact structures (see in Figure 6a–c), and smaller pore volumes. Therefore, the specific surface area of the SDs prepared by spray-drying had no significant increase compared with that of the SDs prepared by vacuum drying.

### 3.7. Dissolution Testing

The dissolution of a poorly water-soluble drug is crucial where it is the rate-limiting step in the oral absorption process from a solid dosage form and is an important parameter related to bioavailability [33,41]. The dissolution profiles of pure AG and SDs are illustrated in Figure 7. It can be seen that the maximum dissolution percentage of pure AG in 120 min was only 22% when using water as the medium (Figure 7).

*Q*_5min_ (the dissolution percentage in 5 min) and *t*_85%_ (time required for 85% dissolution) were calculated and shown in Table 4. As shown in Table 4 and Figure 7, the dissolution rate of AG in the SDs was increased obviously when compared with the pure AG. In the SDs prepared by vacuum-drying method only AG–SM6–V–SD achieved an AG dissolution percentage of 85% under the specified dissolution conditions. In contrast to the SDs prepared by the vacuum-drying method, all SDs prepared by spray-drying method exhibited higher values of *Q*_5min_ and lower values of *t*_85%_. This may be due to the crystallinity of AG in the SDs prepared by spray-drying method was lower than that in the SDs prepared by vacuum-drying method. It was beneficial for the dissolution of AG from SDs prepared by spray-drying method. 

## 4. Discussion

FT-IR (Table 1) showed that the degree of –OH blue shift of AG in AG-grafted PEG–V–SD (except AG–SM3–V–SD and AG–SM6–V–SD) was weaker than that of AG in the corresponding AG-grafted PEG–S–SD. Moreover, the degree of –OH blue shift of AG and the *T*_i_ showed an initial tendency to decrease and then increase on going from AG–SM1–V–SD to AG–SM6–V–SD. A possible reason is that the solvent evaporation rate was slower when SDs were prepared by vacuum-drying method. Therefore, the solidification synchronism of AG and grafted PEG and their intermolecular interaction weakened gradually with the increase of lipophilicity of grafted PEG, resulting in the decrease of the –OH blue shift of AG in AG-grafted PEG–V–SD and the *T*_i_ of AG-grafted PEG–V–SD. It is worth noting that the stronger lipophilicity of SM3 and SM6, especially SM6, weakened the influence of unsynchronized solidification of AG and grafted PEG on their intermolecular interaction, leading to the –OH peak position of AG in AG–SM3–V–SD and AG–SM6–V–SD shifted to 3330 cm^−1^, and the *T*_i_ of AG–SM6–V–SD raised to 323 °C.

DSC results showed that the *T*_m_ of AG–SM1–V–SD and AG–SM6–V–SD was significantly lower than that of AG–SM2/SM3/SM4/SM5–V–SD, which was similar to that of AG-grafted PEG–S–SD. However, XRD results showed that the crystallinity of AG in AG–SM1–V–SD was significantly higher than that in AG–SM1–S–SD (Figure 5b). Therefore, the lower *T*_m_ of AG–SM1–V–SD may be ascribed to the lower melting point of SM1. During heating SM1 melted earlier, allowing the AG to dissolve in earlier melted SM1, thus reducing the crystallinity of AG in AG–SM1–V–SD, not the true crystallinity of AG in AG–SM1–V–SD. The XRD showed that the crystallinity of AG in AG–SM6–V–SD was lower than that of other SDs prepared by vacuum-drying method and was slightly higher than that in AG–SM6–S–SD (Figure 5c). This was because of the higher compatibility of stronger lipophilic SM6 with the poorly water-soluble AG, which slowed down the AG recrystallization caused by the slower solvent evaporation rate during the process of SD prepared by vacuum-drying method. As a result, the crystallinity of AG in AG–SM6–V–SD was reduced to the close degree as that of AG in AG–SM6–S–SD.

The surface morphology of the SDs was strongly affected by the preparation methods and was weakly affected by the properties of the carriers. This may be attributed to the fact that the eight carriers were all PEG and PEG grafted with carbon chains of different lengths, and thus were composed of similar structures with similar physicochemical properties. 

In order to further study the effect of prepared methods and carrier lipophilicity on the dissolution behavior of AG in the SDs, all the samples were investigated by cluster analysis. Based on the variable of the dissolution percentage of AG at different times, the samples were cluster analyzed by intragroup connection clustering method and squared Euclidean distance measured data. The cluster analysis Figure 8 was calculated by SPSS19.0 software ( SPSS, Chicago, USA). 

As can be seen from Figure 8, when the distance was less than 12, the samples can be divided into three groups: the SDs prepared by spray-drying method were the first group; the pure AG were the second group; the SDs prepared by vacuum-drying method were the third group. The results of cluster analysis showed that the dissolution rate of AG could be significantly increased by making the AG and carrier into SDs, and the preparation method had important influences on the dissolution rate of AG in AG–SDs.

When the distance was reduced to less than 2, AG–PEG4000–S–SD was isolated from the first group in a single group. This was consistent with the fact that among the eight carrier materials, the one with the worst lipophilicity, therefore the worst compatibility with AG, was PEG4000. Thus, PEG4000 had the worst effect on improving the dissolution rate of AG.

When the distance was reduced to less than 4, AG–SM6–V–SD was also isolated from the third group. It was suggested that from SM4 to SM6, with the increase of length of carbon chains on grafted PEG8000, the compatibility of the carrier with poorly water-soluble AG was gradually increased. SM6 was more effective than SM4 and SM5 in improving the dissolution behavior of AG in medium. Therefore, the cumulative dissolution percentage of AG in AG–SM6–V–SD at 120 min was up to 92%. 

In the AG-grafted PEG–S–SDs, because of the similar structure and physicochemical properties of six grafted PEGs, and the SDs prepared by spray-drying method can effectively improve the dissolution rate of AG, therefore the effect of grafted PEG lipophilicity on AG dissolution behavior in SD was not observed. However, the SDs prepared by vacuum-drying method had only limited improvement on AG dissolution behavior. Therefore, the influence of the length of carbon chain grafted of PEG8000 on the dissolution behavior of AG in SD can be found. 

## 5. Conclusions

In this work the effect of carrier lipophilicity and preparation method on the physicochemical properties and dissolution behavior of AG–SD were investigated. In the AG–SDs prepared by the different preparation methods with the same polymer as the carrier material, the intermolecular interaction, *T*_i_, *T*_m_, surface morphology, crystallinity, and dissolution behavior had significant differences. Therefore, the preparation method has an important effect on the properties of SD. In the AG–SDs prepared by the same vacuum-drying method with different grafted PEGs as carrier materials, the crystallinity and *T*_m_ decreased, and the dissolution rate of AG increased with the increase of grafted PEG lipophilicity. Therefore, the increase of carrier material lipophilicity is beneficial to the thermal stability of SD, the decrease of crystallinity, and the increase of the dissolution rate of a poorly water-soluble drug in the SD. 

## Figures and Tables

**Figure 1 pharmaceutics-11-00074-f001:**
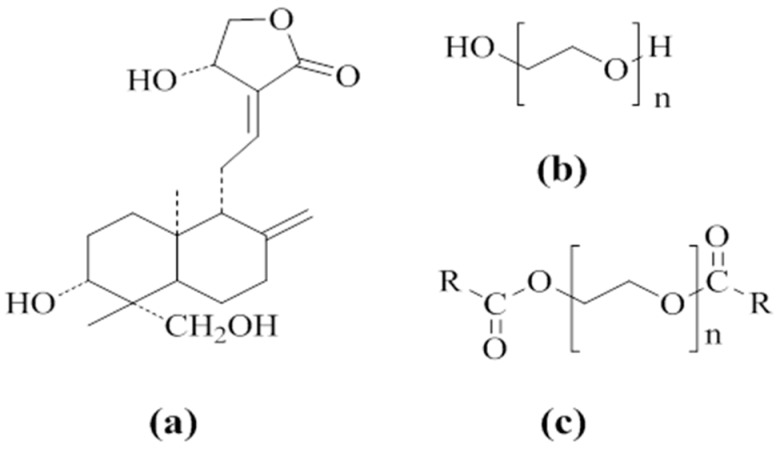
Molecular structures of andrographolide (AG) (**a**), polyethylene glycol (PEG) (**b**), and PEG grafted with carbon chains (**c**): R = C_11_H_23_ in PEG4000 laurate (SM1) and PEG8000 laurate (SM4), R = C_15_H_31_ in PEG4000 palmitate (SM2) and PEG8000 palmitate (SM5), R = C_22_H_45_ in PEG4000 behenate (SM3) and PEG8000 behenate (SM6).

**Figure 2 pharmaceutics-11-00074-f002:**
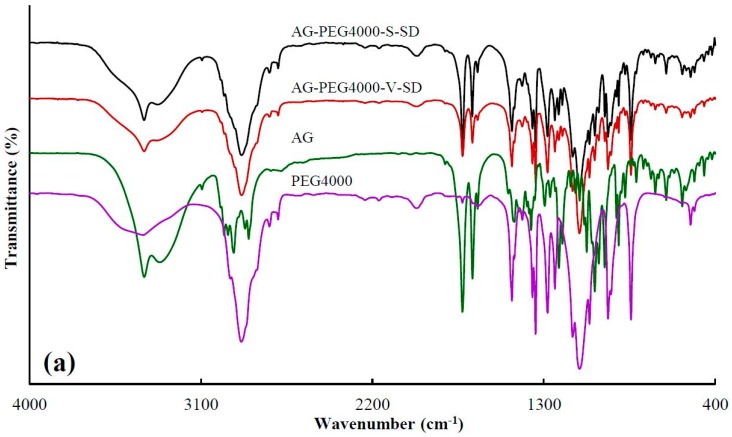
FT-IR spectra of pure andrographolide (AG), carrier materials, and solid dispersions (SD): (**a**) PEG4000 as carrier material, (**b**) SM1 as carrier material, (**c**) SM6 as carrier material.

**Figure 3 pharmaceutics-11-00074-f003:**
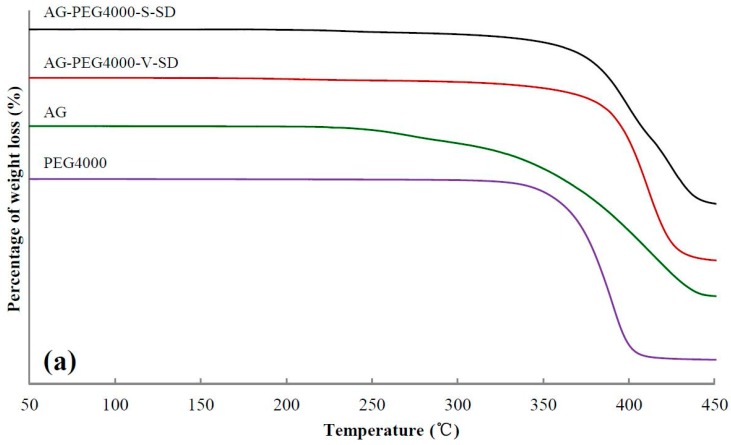
Thermogravimetric (TG) curves of pure andrographolide (AG), carrier materials, physical mixtures (PM), and solid dispersions (SD): (**a**) PEG4000 as carrier material, (**b**) SM1 as carrier material, (**c**) SM6 as carrier material.

**Figure 4 pharmaceutics-11-00074-f004:**
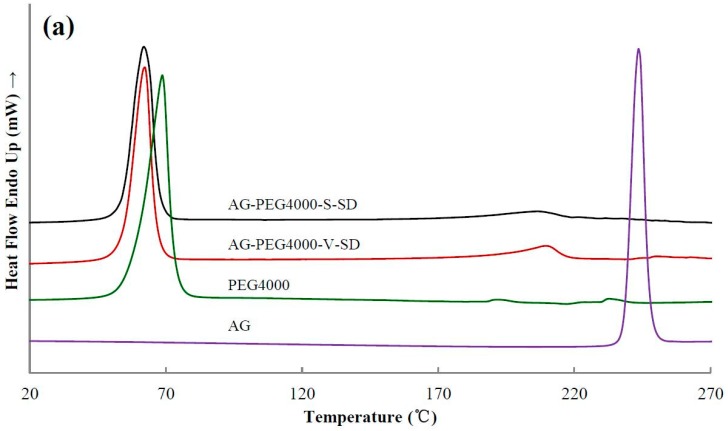
Differential scanning calorimetry (DSC) thermograms of pure andrographolide (AG), carrier materials, and solid dispersions (SD): (**a**) PEG4000 as carrier material, (**b**) SM1 as carrier material, (**c**) SM6 as carrier material.

**Figure 5 pharmaceutics-11-00074-f005:**
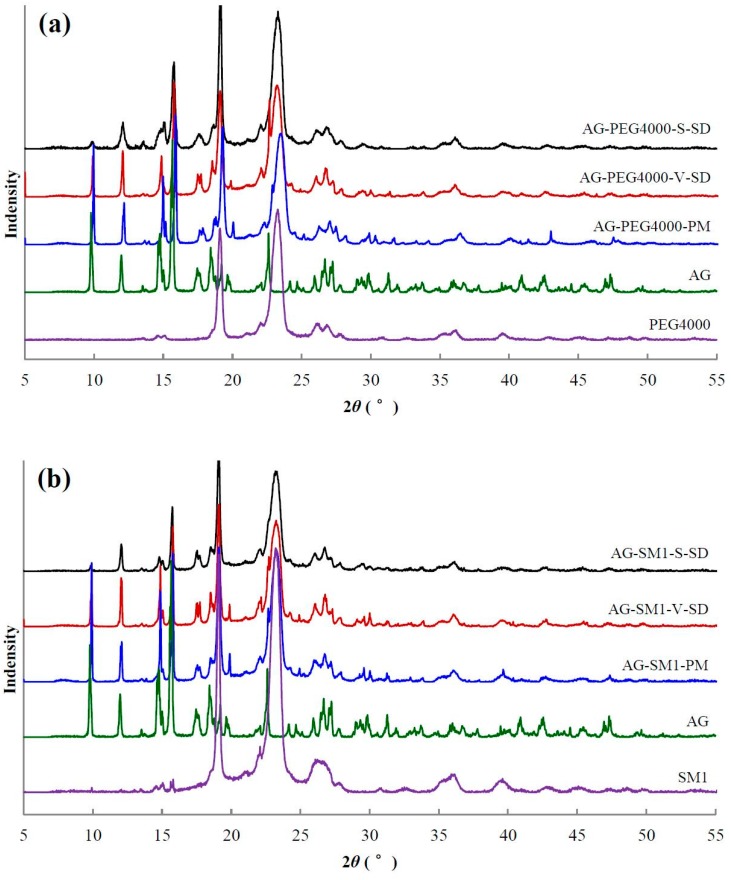
X-ray diffractogram of pure andrographolide (AG), carrier materials, physical mixtures (PM), and solid dispersions (SD): (**a**) PEG4000 as carrier material, (**b**) SM1 as carrier material, (**c**) SM6 as carrier material.

**Figure 6 pharmaceutics-11-00074-f006:**
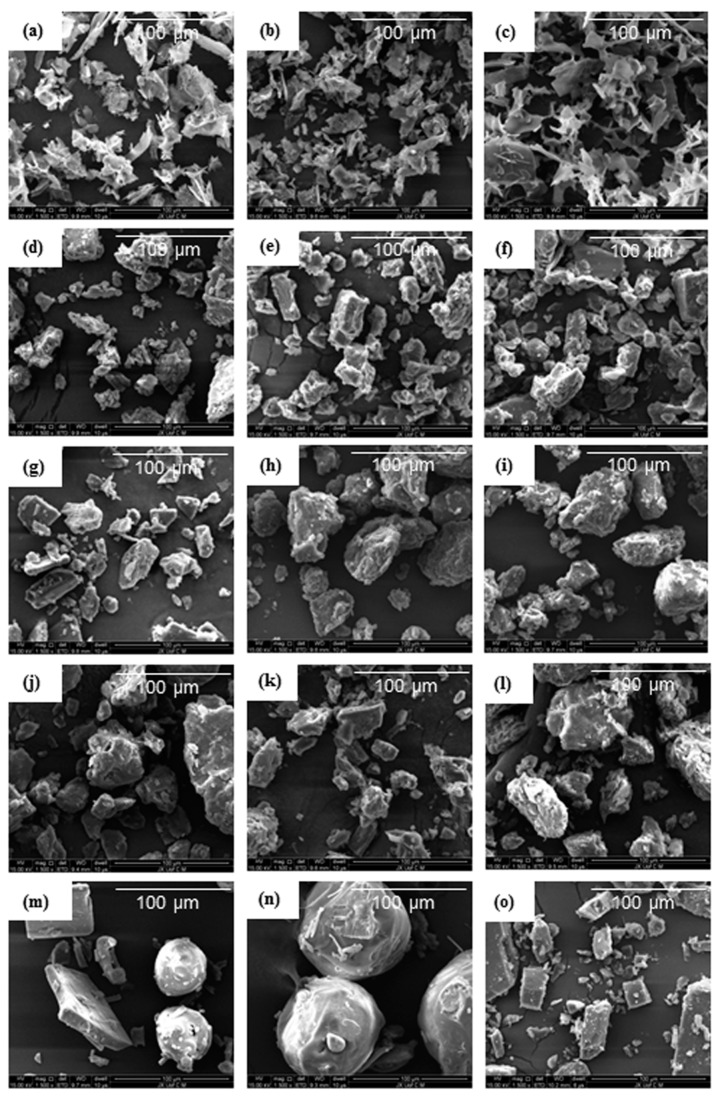
Scanning Electron Microscopy photographs of pure andrographolide (AG), carrier materials (PEG4000, PEG8000, SM1, and SM6), physical mixtures (PM), and solid dispersions (SD): (**a**) AG–PEG4000–S–SD, (**b**) AG–SM1–S–SD, (**c**) AG–SM6–S–SD, (**d**) AG–PEG4000–V–SD, (**e**) AG–SM1–V–SD, (**f**)AG–SM6–V–SD, (**g**) AG–PEG4000–PM, (**h**) AG–SM1–PM, (**i**) AG–SM6–PM, (**j**) PEG4000, (**k**) SM1, (**l**) SM6, (**m**) AG–PEG8000–PM, (**n**) PEG8000, (**o**) AG.

**Figure 7 pharmaceutics-11-00074-f007:**
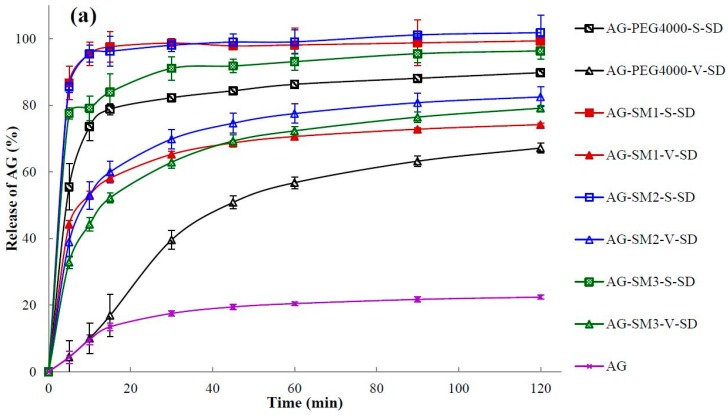
Dissolution profiles of pure andrographolide (AG) and solid dispersions (SD): (**a**) PEG4000, SM1, SM2, and SM3 as carrier materials, (**b**) PEG8000, SM4, SM5, and SM6 as carrier materials (*n* = 6).

**Figure 8 pharmaceutics-11-00074-f008:**
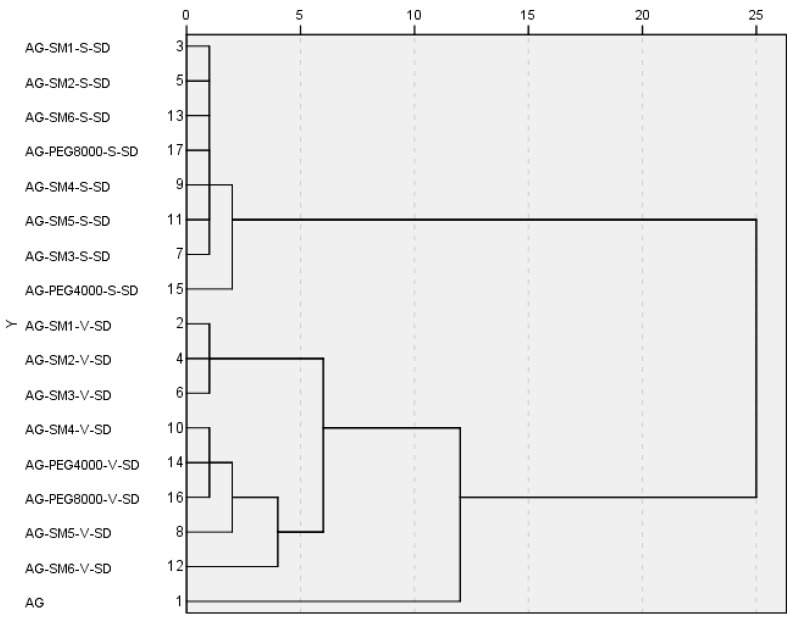
Cluster analysis tree diagram by intergroup connection on the variable of the dissolution percentage at different times.

**Table 1 pharmaceutics-11-00074-t001:** –OH peak position of andrographolide in pure andrographolide (AG), and solid dispersion (SD). [PEG4000 laurate (SM1), PEG4000 palmitate (SM2), PEG4000 behenate (SM3), PEG8000 laurate (SM4), PEG8000 palmitate (SM5), and PEG8000 behenate (SM6)].

Sample	Peak Position (cm^−1^)	Sample	Peak Position (cm^−1^)
AG	3317.0 ± 1.4		
AG–PEG4000–S–SD	3323.9 ± 0.9	AG–PEG4000–V–SD	3332.4 ± 1.7
AG–PEG8000–S–SD	3321.0 ± 1.0	AG–PEG8000–V–SD	3331.3 ± 0.5
AG–SM1–S–SD	3328.3 ± 0.4	AG–SM1–V–SD	3327.4 ± 0.8
AG–SM2–S–SD	3328.7 ± 1.3	AG–SM2–V–SD	3324.3 ± 0.8
AG–SM3–S–SD	3322.3 ± 1.6	AG–SM3–V–SD	3329.9 ± 1.3
AG–SM4–S–SD	3327.7 ± 0.3	AG–SM4–V–SD	3322.7 ± 2.5
AG–SM5–S–SD	3328.4 ± 1.3	AG–SM5–V–SD	3325.4 ± 1.2
AG–SM6–S–SD	3329.1 ± 0.2	AG–SM6–V–SD	3330.4 ± 2.6

**Table 2 pharmaceutics-11-00074-t002:** *T*_i_ (5% weight-loss temperature) and *T*_m_ (melting temperature) of pure andrographolide (AG), physical mixture (PM), and solid dispersion (SD).

Sample	*T*_i_(°C)	*T*_m_(°C)	Sample	*T*_i_(°C)	*T*_m_(°C)	Sample	*T*_i_(°C)
AG	272	243					
AG–PEG4000 –S–SD	335	206	AG–PEG4000 –V–SD	345	210	AG–PEG4000 –PM	346
AG–PEG8000 –S–SD	336	209	AG–PEG8000 –V–SD	348	217	AG–PEG8000 –PM	345
AG–SM1–S–SD	298	220	AG–SM1–V–SD	318	220	AG–SM1–PM	304
AG–SM2–S–SD	297	221	AG–SM2–V–SD	317	229	AG–SM2–PM	318
AG–SM3–S–SD	319	221	AG–SM3–V–SD	304	230	AG–SM3–PM	317
AG–SM4–S–SD	311	218	AG–SM4–V–SD	308	231	AG–SM4–PM	329
AG–SM5–S–SD	327	218	AG–SM5–V–SD	305	227	AG–SM5–PM	336
AG–SM6–S–SD	308	219	AG–SM6–V–SD	323	221	AG–SM6–PM	323

**Table 3 pharmaceutics-11-00074-t003:** Specific surface area, pore volume, particle size, and distribution of solid dispersions.

Sample	Specific Surface Area (m^2^/g)	Pore Volume (×10^3^, m^3^/g)	d0.5 (μm)	Span	Sample	Specific Surface Area (m^2^/g)	Pore Volume (×10^3^, m^3^/g)	d0.5 (μm)	Span
AG–PEG4000 –S–SD	0.5641	0.964	42.8	2.748	AG–PEG4000 –V–SD	0.2969	1.650	109.4	7.036
AG–PEG8000 –S–SD	0.1943	0.148	33.1	4.524	AG–PEG8000 –V–SD	0.9994	1.745	113.3	4.425
AG–SM1–S–SD	0.5871	0.689	52.7	2.538	AG–SM1–V–SD	0.3739	3.546	131.3	6.747
AG–SM2–S–SD	0.2857	0.932	79.6	3.920	AG–SM2–V–SD	0.2765	1.761	96.0	8.696
AG–SM3–S–SD	0.1126	0.656	64.3	2.946	AG–SM3–V–SD	0.0337	3.405	125.4	5.621
AG–SM4–S–SD	0.3863	1.094	48.6	3.302	AG–SM4–V–SD	0.2478	3.411	127.2	5.198
AG–SM5–S–SD	0.3056	0.965	52.4	2.813	AG–SM5–V–SD	0.6194	1.542	99.8	5.199
AG–SM6–S–SD	0.5387	0.957	48.5	2.158	AG–SM6–V–SD	0.4387	2.160	144.7	5.020

**Table 4 pharmaceutics-11-00074-t004:** *Q*_5min_ and *t*_85%_ of pure andrographolide (AG), physical mixtures (PM), and solid dispersions (SD).

Sample	*Q*_5min_ (%)	*t*_85%_ (min)	Sample	*Q*_5min_ (%)	*t*_85%_ (min)
AG	4.32 ± 1.84	>120			
AG–PEG4000–S–SD	55.49 ± 6.95	50	AG–PEG4000–V–SD	4.37 ± 2.50	>120
AG–PEG8000–S–SD	82.86 ± 3.90	6	AG–PEG8000–V–SD	3.86 ± 1.90	>120
AG–SM1–S–SD	86.72 ± 5.10	5	AG–SM1–V–SD	44.22 ± 1.15	>120
AG–SM2–S–SD	85.61 ± 1.85	5	AG–SM2–V–SD	38.97 ± 4.44	>120
AG–SM3–S–SD	77.57 ± 1.66	17	AG–SM3–V–SD	32.85 ± 1.80	>120
AG–SM4–S–SD	80.24 ± 2.31	7	AG–SM4–V–SD	19.64 ± 0.75	>120
AG–SM5–S–SD	87.97 ± 4.02	5	AG–SM5–V–SD	22.17 ± 1.87	>120
AG–SM6–S–SD	79.40 ± 4.35	6	AG–SM6–V–SD	19.40 ± 1.31	67

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
