# Peer review of "Effect of Carrier Lipophilicity and Preparation Method on the Properties of Andrographolide–Solid Dispersion"

_pharmaceutics, 2019, doi:10.3390/pharmaceutics11020074_

Reviewer 1 Report

Please see the attached document for the comments.

Author Response

Dear Sir./Madam:

    We thank you for your careful reading of the manuscript and helpful comments and suggestions. We have made revisions according to your comments, as described below Word.

Yashao Chen

Reviewer 2 Report

This paper reports the successful improvement of the dissolution rate of an API by SD formation. I have just two points to make.

Could the authors add an XRPD particle size determination based on peak widths? 

The quality of the English language used is quite good but there is a minor recurring fault in the use of singluar when plural should be used, e.g. Page 1 line 22 "In the AG-SDs prepared by the different preparation method" "methods" would be better.  

Author Response

Dear Sir./Madam:

    We thank you for your careful reading of the manuscript and helpful comments and suggestions. We have made revisions according to your comments, as described below Word.

Yashao Chen

Round  2

Reviewer 1 Report

I continue to have issues with the notion of  defining the dispersions in the current manuscript. There is a need to choose the right control for both the vacuum and the spray drying products. There is no significant difference between the performance of the PEG and the grafted polymers which according to me is due to the drug : polymer ratio to start with. It is expected that a composition with lower proportion of the API is thermodynamically stable. I would recommend 15%  drug loading as a control. Starting with this composition and then introducing grafting with varying lipophilicities would be beneficial in terms of the stabilizing effect (physical stability). There is no mention of the solubility enhancement achieved via solid dispersion.

            It is very difficult to comprehend Response 6. As the glass transition of andrographolide is 75, the stability of the dispersions would differ with the operating temperature of the vacuum dryer. Provide further evidence (any DSC, PXRD data for the products obtained at those three temperatures) or literature support.

            With respect to response 21, did the authors sample after performing an infinite spin at the end of dissolution ? At t=120 minutes, was the entire drug solubilized in case of most of the samples ? This would impact all the calculations.

     It is imperative to show physical stability of these dispersions else performance may not be of substantial value.

Author Response

Dear Sir. /Madam:

   We thank you for your careful reading of the manuscript and helpful comments and suggestions. We have made revisions according to your comments, as described below Word.

Yashao Chen

Round  3

Reviewer 1 Report

N/A